# Cholesterol mediated ferroptosis suppression reveals essential roles of Coenzyme Q and squalene

Qi Sun[1,5], Diming Liu[1,5], Weiwei Cui[2], Huimin Cheng[1], Lixia Huang[1], Ruihao Zhang[1], Junlian Gu[3], Shuo Liu[4], Xiao Zhuang[2], Yi Lu [1✉], Bo Chu [2✉] & Jian Li [1✉]

Recent findings have shown that fatty acid metabolism is profoundly involved in ferroptosis. However, the role of cholesterol in this process remains incompletely understood. In this work, we show that modulating cholesterol levels changes vulnerability of cells to ferroptosis. Cholesterol alters metabolic flux of the mevalonate pathway by promoting Squalene Epoxidase (SQLE) degradation, a rate limiting step in cholesterol biosynthesis, thereby increasing both CoQ10 and squalene levels. Importantly, whereas inactivation of Farnesyl-Diphosphate Farnesyltransferase 1 (FDFT1), the branch point of cholesterol biosynthesis pathway, exhibits minimal effect on ferroptosis, simultaneous inhibition of both CoQ10 and squalene biosynthesis completely abrogates the effect of cholesterol. Mouse models of ischemia-reperfusion and doxorubicin induced hepatoxicity confirm the protective role of cholesterol in ferroptosis. Our study elucidates a potential role of ferroptosis in diseases related to dysregulation of cholesterol metabolism and suggests a possible therapeutic target that involves ferroptotic cell death.

[1] Department of Biochemistry and Molecular Biology, School of Basic Medical Sciences, Cheeloo College of Medicine, Shandong University, Jinan, Shandong 250012, China. [2] Department of Cell Biology, School of Basic Medical Sciences, Cheeloo College of Medicine, Shandong University, Jinan, Shandong 250012, China. [3] School of Nursing and Rehabilitation, Cheeloo College of Medicine, Shandong University, Jinan, Shandong 250012, China. [4] Department of Geriatric Medicine, Qilu Hospital, Shandong University, Wenhuaxi Road 107, Jinan, Shandong 250012, China. [5]These authors contributed equally: Qi Sun, Diming Liu. ✉email: luyi@sdu.edu.cn; chubo123@sdu.edu.cn; jianli08@sdu.edu.cn

The essentiality of cholesterol for mammals is revealed by its indispensable role in membrane fluidity and permeability control, as well as serving as precursor for biosynthesis of bile acids and steroid hormones. Cholesterol dyshomeostasis has been shown to be involved in an array of devastating diseases. On the one hand, high levels of cholesterol are linked to atherosclerosis[1], NAFLD/NASH[2], obesity/diabetes[3] and cancer[4]. On the other hand, hypocholesterolemia has a profound association with higher morbidity and motility from clinically ill patients[5–7]. Due to its importance in normal functions of cells and the whole body, cholesterol metabolism is strictly controlled. Cells maintain their cholesterol levels mainly through dynamic balance between LDL uptake, esterification of excess intracellular cholesterol, cholesterol excretion and biosynthesis.

Starting from acetyl-CoA, the mevalonate pathway is responsible for endogenous cholesterol biosynthesis catalyzed by 22 enzymes[8]. Importantly, in addition to cholesterol, this pathway also produces a few sterol intermediates, including lanosterol, 7-dehydrocholesterol (7-DHC) and desmosterol, as well as nonsterol molecules such as dolichol, heme A, farnesyl, geranylgeranyl, isopentenyl pyrophosphate (IPP) and coenzyme Q (CoQ). Cholesterol and some sterol metabolites in the mevalonate pathway play regulatory roles in cholesterol metabolism. For example, excessive membrane cholesterol in the endoplasmic reticulum (ER) induces conformational change of squalene epoxidase (SQLE) and subsequent degradation mediated by MARCH6[9,10]. 7-DHC, the direct precursor to cholesterol in the Kandutsch-Russell branch of cholesterol biosynthesis pathway, accumulates in Smith-Lemli-Opitz syndrome (SLOS) and accelerates proteolysis of HMG-CoA reductase to suppress sterol biosynthesis[11]. Desmosterol build-up in macrophage foam cells activates Liver X receptor (LXR) and inhibits SREBP dependent gene transcription to suppress inflammatory response[12]. Recent years have witnessed novel functions of sterols beyond their roles in cholesterol metabolism. Virus infection was reported to reduce 7-dehydrocholesterol reductase (DHCR7) expression, the resultant higher levels of 7-DHC enhanced type I interferon production and promoted virus clearance in macrophages[13]. Very recently, 7-DHC has also been shown to have superior reactivity toward peroxyl radicals and effectively protect phospholipids from peroxidation and ferroptotic cell death in cancer cells[14].

Ferroptosis is a newly identified programmed cell death characterized by massive lipid peroxidation of cell membranes[15]. Advances in the mechanistic studies have highlighted an intersection of ferroptosis with metabolic regulation[16], including energy metabolism[17], fatty acid metabolism[18], amino acid metabolism[19] and cholesterol metabolism. Several intermediates synthesized from the mevalonate pathway modulate sensitivity of cells to ferroptosis. For example, IPP is critical for maturation of GPX4[20], the first reported protein to protect cells from ferroptosis. Coenzyme Q and squalene, two other hydrophobic metabolites, have been implicated to act as novel lipid peroxide scavengers besides GPX4[21–23]. Additionally, chronic exposure of cancer cells to 27-hydroxycholesterol, a derivative of cholesterol, selects for more metastatic cancer cells that increases resistance to ferroptosis[24]. Like other unsaturated phospholipid, cholesterol is intrinsically susceptible to peroxidation upon oxidative stress, which has been suggested to lead to damage of membrane integrity and function[25]. Nevertheless, whether and how cholesterol contributes to plasticity of cells to ferroptosis remains incompletely known.

Since the mevalonate pathway is closely related to ferroptosis through regulating expression of GPX4, production of squalene and coenzyme Q, and sterols are known to exert intricate feed-back regulations on this pathway to restrict cholesterol overproduction, we hypothesized that cholesterol and its metabolites would modulate cell sensitivity to ferroptosis. To this end, we performed a small-scale screening of commercially available sterol metabolites/derivatives as potential modulators of ferroptosis, which include those that influence SREBP/SCAP transport from the endoplasmic reticulum to Golgi (25- and 27- hydroxycholesterol, denoted as 25-HC and 27-HC respectively), intermediates of cholesterol biosynthesis that control protein stabilities of HMGCR and SQLE, two rate-limiting enzymes for cholesterol biosynthesis (desmosterol (Desmo), cholesterol (CH), lanosterol (Lano) and 24, 25-dihydrolanosterol(24,25D-Lano))[26], direct precursor to cholesterol (7-DHC), as well as other sterols/oxysterols that may regulate cholesterol metabolism (zymosterol (Zymo) and lathosterol (Lano)). In this screen, we finally identified cholesterol and desmosterol (hereafter referred as CH&Desmo) as endogenous ferroptosis inhibitors. We found that these two metabolites in the mevalonate pathway exerted their anti-ferroptotic functions through the same mechanism: accelerating degradation of SQLE to increase intracellular levels of both squalene and coenzyme Q. Through a series of genetic and pharmacological approaches, we validated the critical roles of these two known lipid peroxide scavengers in the action of CH&Desmo. More importantly, data in mouse models confirmed that cholesterol feeding alleviated both ischemia-reperfusion and doxorubicin induced hepatotoxicity through blocking ferroptosis. Our findings unraveled cholesterol as a protective factor in tissue damage and suggested "the other side of the coin" for cholesterol, especially regarding overwhelming studies reporting diseases associated with cholesterol overload.

## Results

**Metabolites of cholesterol biosynthesis pathway inhibit ferroptosis.** To explore the role of cholesterol metabolism in ferroptosis, we tested the effect of a variety of metabolites in cholesterol biosynthetic pathway on ferroptosis. HT1080 cells were first pretreated with these metabolites for 3 h to allow uptake, after thorough washout, ferroptosis was induced by addition of RSL3, a GPX4 inhibitor. This initial screen identified desmosterol (Desmo), cholesterol (CH) and 7-dehydrocholesterol (7-DHC) to significantly protect HT1080 cells from RSL3 induced lipid peroxidation and ferroptosis. Among these three molecules, 7-DHC showed the most potent effect, with a near complete inhibition achieved at 40 μM (Fig. 1a–c and Supplementary Fig. 1a, c). This result was further validated in 786-O cells (Supplementary Fig. 1b). 7-DHC and desmosterol are two immediate precursors to cholesterol in the Kandutsch-Russell and Bloch branch of cholesterol biosynthetic pathway, respectively (Supplementary Fig. 1d)[27], with the former shown as an endogenous ferroptosis suppressor by directly inhibiting lipid peroxidation[14]. We thus focused our study on desmosterol and cholesterol hereafter.

To characterize the protective functions of desmosterol and cholesterol in ferroptosis in more detail, we utilized a couple of well-established approaches to induce ferroptotic cell death. Cysteine starvation and erastin mediated inhibition of cystine uptake lead to decreased glutathione synthesis, which provides reducing power for GPX4, a key anti-ferroptotic protein. In line with the RSL3 data, cholesterol rendered HT1080 cells more resistant to RSL3, cysteine starvation, erastin treatment as well as when *GPX4* is deleted by CRISPR mediated knockout, with a comparable potency to Fer-1 (ferrostatin-1) (Fig. 1d–h). Similar results were obtained with desmosterol (Fig. 1i–l). As a further confirmation that these effects were not restricted to one specific cell type, CH&Desmo also markedly mitigated ferroptosis in

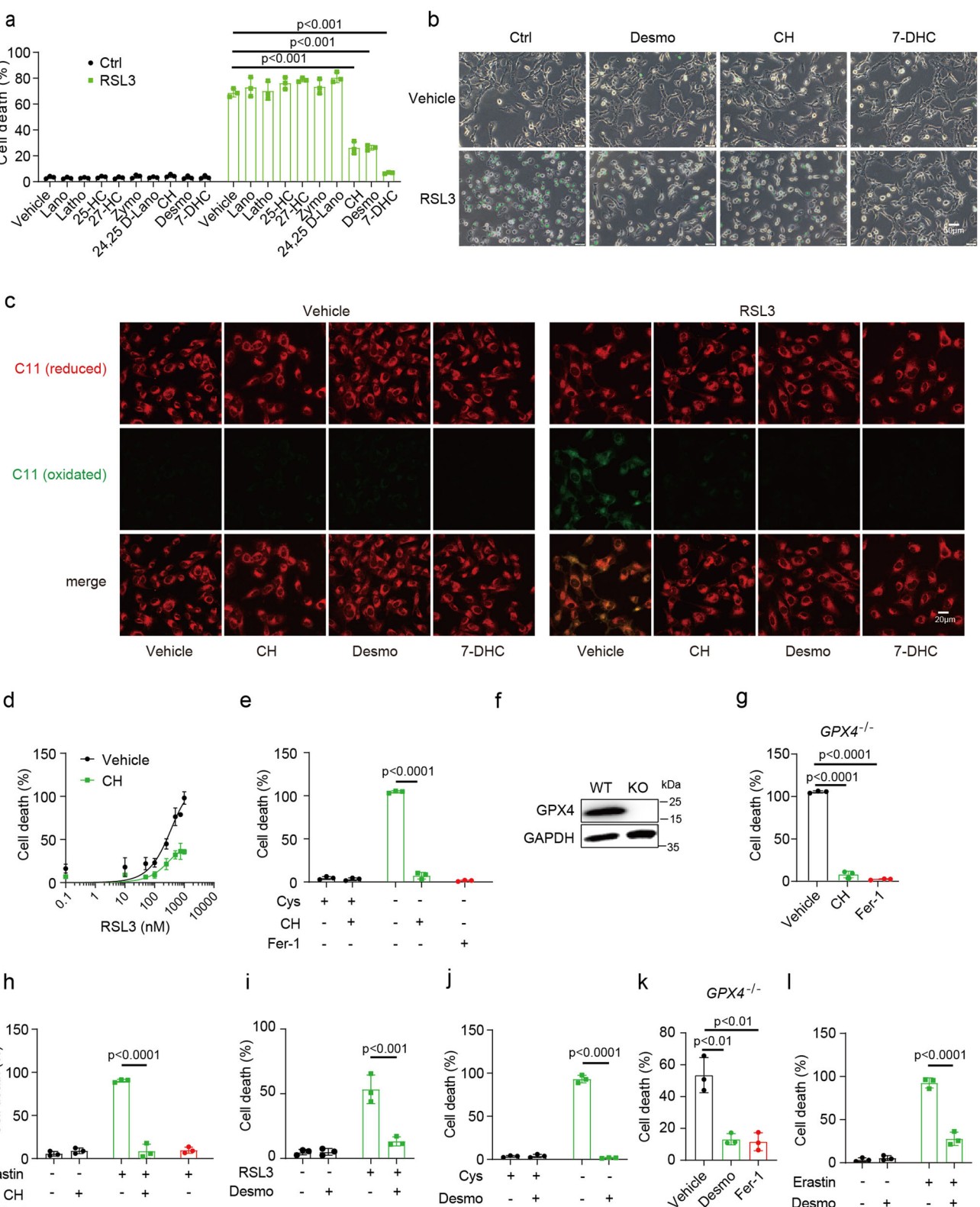

786-O cells (Supplementary Fig. 1e–g for cholesterol, Supplementary Fig. 1h–j for desmosterol). Interestingly, time course experiments revealed that CH&Desmo gradually lost their activities, with the latter remaining about 50% inhibition after 9 h washout. In contrast, 7-DHC almost completely inhibited ferroptosis even after 24 h, suggesting a different mechanism or metabolic kinetics with CH&Desmo (Supplementary Fig. 1k–m).

Further dose-dependent analysis indicated that CH&Desmo exhibited similar efficacy on suppression of ferroptosis, both starting to show a trend of inhibition at 20 μM, and 40 μM was sufficient to almost entirely abolish RSL3-induced cell death (Supplementary Fig. 2a, b). Taken together, these data suggest that cholesterol or desmosterol is able to substantially attenuate ferroptotic cell death.

**Fig. 1 Metabolites of cholesterol biosynthesis pathway inhibit ferroptosis. a** Identification of sterol metabolites as ferroptosis inhibitors in HT1080 cells. Cells were pretreated with metabolites for 3 h, drugs were washed out with serum-free medium before cells were treated with RSL3 (500 nM) for 4 h. Desmo, desmosterol, 40 μM; Lano, Lanosterol, 40 μM; Latho, Lathosterol, 40 μM; 25-HC, 25-hydroxycholesterol, 25 μM; CH, cholesterol, 40 μM; 7-DHC, 7-dehydrocholesterol, 40 μM; Zymo, zymosterol, 40 μM; 27-HC, 27- hydroxycholesterol, 40 μM; 24,25 D-Lano, 24, 25-dihydrolanosterol, 40 μM. Dead cells were stained with SYTOXGreen. **b** Representative phase-contrast images of HT1080 cells in (**a**). Scale bar, 50 μm. **c** Immunofluorescence staining with BODIPY- C11 581/591 to dectect the levels of lipid peroxidation in cells treated with CH (40 μM), Desmo(40 μM) and 7-DHC (40 μM). Scale bar, 20 μm. **d** Dose-dependent toxicity of RSL3 in HT1080 cells with or without cholesterol (40 μM) treatment. **e** Cell death measurement of HT1080 cells co-treated with cystine-free medium and cholesterol (40 μM) for 20 h, Fer-1, 1 μM ferrostatin-1. **f** Immunoblot detection of GPX4 protein for wildtype and $GPX4^{-/-}$ HT1080 cells. **g** Cell death measurement of HT1080 $GPX4^{-/-}$ cells treated with CH (40 μM) or Fer-1 (1 μM). Cells were maintained in the presence of Fer-1. After supplementing MβCD/cholesterol for 3 h, Fer-1 was then removed to induce ferroptosis. **h** HT1080 cells were pre-incubated with 5 μM Erastin for 12 h, followed by CH (40 μM) for 3 h. After thorough washout, cells were continued to be treated with Erastin for 3 h. **i–l** Cell death analysis of HT1080 cells treated with desmosterol (40 μM), together with RSL3 (500 nM) (**i**), or with cystine-free medium (**j**), or in $GPX4^{-/-}$ cells (**k**) or with Erastin (5 μM) (**l**). Culture conditions are the same with that of cholesterol treatment. Dead cells were stained with SYTOX Green (**a**, **d–l**). Data are presented as mean ± SD, n = 3 (**a**, **d–l**) independent repeats. Significance in (**d**, **e**, **h**, **i**, **j**, **l**) was calculated using two-tailed unpaired Student's t-test. Significance in (**a**, **g**, **k**) was calculated using a one-way ANOVA with Tukey's post hoc test.

**Functional characterization of cholesterol and desmosterol**. We asked whether perturbation of cholesterol metabolism from different aspects also influenced cell sensitivity to ferroptosis. As an alternative approach to increase cellular cholesterol content, cells were pretreated with avasimibe for 12 h (Supplementary Fig. 2d), followed by ferroptosis induction with RSL3. Avasimibe is a specific inhibitor of acyl coenzyme A-cholesterol acyltransferase (ACAT) that catalyzes esterification of free cholesterol. Avasimibe treatment increased cell death resistance to RSL3 (Fig. 2a) and cysteine starvation (Supplementary Fig. 2c), corroborating our findings in exogenously added cholesterol. Methyl-β-cyclodextrin (MβCD) has been widely used to extract cholesterol from cell plasma membrane. In contrast to avasimibe treatment, depletion of cellular cholesterol with 1% MβCD for 1 h (Supplementary Fig. 2e) promoted sensitivity of HT1080 cells to both RSL3 (Fig. 2b) and cysteine starvation (Supplementary Fig. 2f) induced ferroptosis. Of note, no cell viability was altered with MβCD treatment alone. This result was further verified by inhibition of NPC1, a lysosomal cholesterol transporter mediating uptake of LDL derived cholesterol, with U18666A (Fig. 2c and Supplementary Fig. 2g, h).

Increased lipid peroxidation is a hallmark of ferroptosis. We next assessed whether CH&Desmo reduced lipid peroxidation during ferroptosis. Indeed, high levels of lipid peroxidation were detected after 2 h of RSL3 supplementation, as monitored by flow cytometry using BODIPY-C11 581/591 staining, which were significantly scavenged by CH&Desmo (Fig. 2d, e). As upstream precursors to cholesterol, excess intermediates of cholesterol biosynthesis may enhance cholesterol biosynthesis, we thus tested this possibility by measuring intracellular cholesterol levels in the presence of sterol intermediates. Desmo and 7-DHC significantly increased cellular cholesterol levels as measured by flow cytometry, slightly less than that of CH supplementation. Latho also induced a marginal increase of intracellular cholesterol levels, whereas Zymo and Lano had no effect (Supplementary Fig. 2i). However, the anti-ferroptotic activity of desmosterol was not impaired by pharmacological inhibition of DHCR24 with triparanol (Fig. 2f, Supplementary Fig. 1d). Notably, like other cholesterol biosynthesis inhibitors, triparanol activated transcription of cholesterol biosynthetic genes SQLE and FDFT1, indicating a functional consequence of DHCR24 inhibition (Supplementary Fig. 2j). We confirmed this result by CRISPR-CAS9 mediated DHCR24 deletion (Fig. 2g, h). In the meantime, excess cholesterol has also been shown to result in higher cellular desmosterol level[12]. To dissect a possible dependency of cholesterol on desmosterol to suppress ferroptosis, we used AY9944 to inhibit DHCR7. DHCR7 catalyzes the formation of desmosterol from 7-dehydrodesmosterol in the cholesterol

biosynthesis pathway. Like triparanol, AY9944 also activated a transcriptional program of cholesterologenesis, as exemplified by increased mRNA levels of SQLE and FDFT1 (Supplementary Fig. 2k). Again, no antagonism between AY9944 and cholesterol was detected (Fig. 2i). As a complementary approach, experiments were also performed in $DHCR7^{-/-}$ cells with the same conclusion (Fig. 2j–l). Thus, CH&Desmo block ferroptotic cell death independently of each other.

**Cholesterol and desmosterol promote ferroptosis resistance partly through FSP1-CoQ10 axis**. We sought to explore the mechanism by which CH&Desmo contributed to ferroptosis resistance. High cellular free cholesterol levels may lead to enhanced lipogenesis through LXR-SREBP1 axis. However, LXR agonist GW3965 increased the expression of LXR target gene ABCA1 without affecting RSL3 induced ferroptosis (Supplementary Fig. 3a, b). In addition, in vitro lipid peroxidation assay using STY-BODIPY as an indicator suggested that, on the contrary to 7-DHC, CH&Desmo unlikely directly reduced phospholipid peroxidation (Supplementary Fig. 3c, d). We then examined the expression levels of known ferroptosis associated proteins. Western blotting indicated that neither the protein expression of ACSL4, GPX4-dependent pathway (GPX4, GCLM and SLC7A11) (Supplementary Fig. 3e-h), nor that of NRF2 targets (HMOX and NQO1) (Supplementary Fig. 3g, h) were altered upon cholesterol or desmosterol treatment.

The mevalonate pathway, which synthesizes farnesyl pyrophosphate to generate CoQ10 or cholesterol, is subjected to feedback regulation by cholesterol. We reasoned that disturbing cholesterol metabolism may concomitantly influence CoQ10 synthesis, which has been reported to sequester lipid peroxidation in GPX4 independent manner. As expected, mass spectrometry assay indicated that, whereas inhibition of CoQ10 biosynthesis enzyme COQ2 with 4-chlorobenzoic acid (4-CBA) reduced, CH&Desmo significantly increased cellular CoQ10 levels (Fig. 3a, b). To investigate whether CoQ10 was involved in the action of CH&Desmo, cells were preincubated with 4-CBA to deplete CoQ10. Supplementary Fig. 3i shows that 4-CBA strongly reduced cell viability in the presence of RSL3. More importantly, the protective effects of CH&Desmo on ferroptosis were largely attenuated by 4-CBA (Fig. 3c, d). The results in stable CoQ2 knockdown cell lines were consistent with what was observed in 4-CBA treated cells (Fig. 3e, f, Supplementary Fig. 3j). Furthermore, addition of 250 nM exogenous CoQ10 led to around 25% increase of intracellular CoQ10 content as measured by mass-spectrometry assay (Supplementary Fig. 3k). Importantly, it also reduced RSL3 induced cell death (Fig. 3k), suggesting that increasing cellular

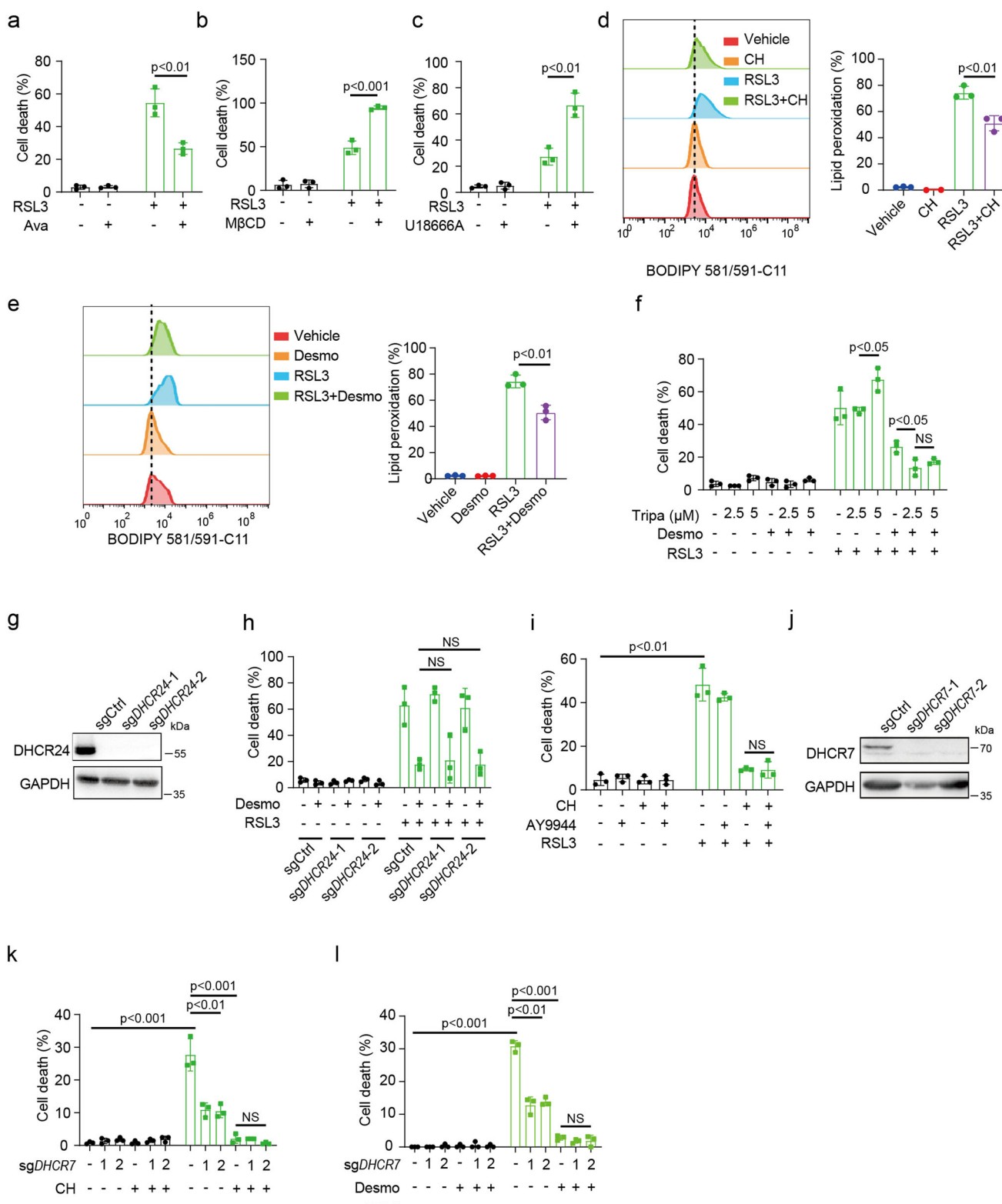

CoQ10 content can inhibit ferroptosis. CoQ can either be reduced by mitochondrial DHODH or cytosolic FSP1 to produce CoQH2, a radical-trapping antioxidant with anti-ferroptosis activity. However, no changes of protein expression of DHODH was observed by cholesterol treatment (Supplementary Fig. 3l). Moreover, inhibition of DHODH by brequinar (BQR) did not obviously affect functions of cholesterol (Fig. 3g) and desmosterol (Supplementary Fig. 3m). This prompted us to

analyze whether FSP1-CoQ10 axis was involved. To this end, we generated $FSP1^{-/-}$ HT1080 cells through CRISPR-CAS9 mediated gene knockout (Fig. 3h). Interestingly, deletion of FSP1 remarkably abrogated CH&Desmo induced ferroptosis resistance (Fig. 3i, j), to a similar extent with 4-CBA treatment (Fig. 3c, d). Altogether, these data suggest that CH&Desmo reduce susceptibility of cells to ferroptosis partly through FSP1-CoQ10 pathway.

**Fig. 2 Functional characterization of cholesterol and desmosterol. a** Cell death measurement of HT1080 cells pre-incubated with avasimible (10 μM) for 12 h followed by RSL3 (500 nM) for 4 h. **b** Cell death measurement of HT1080 cells pre-incubated with 1% MβCD for 30 min followed by RSL3 (500 nM) for 5 h. **c** Cell death measurement of HT1080 cells pre-incubated with U18666A (5 μM) for 12 h followed by RSL3 (500 nM) for 4 h. **d, e** Lipid peroxidation assay of HT1080 cells treated with RSL3. Cells were pretreated with cholesterol (40 μM) (**d**) or desmosterol (40 μM) (**e**) for 3 h before washout, then cells were treated with RSL3 (250 nM) for 2 h. Lipid peroxidation was assessed by flow cytometry. **f** Cell death measurement of HT1080 cells induced by RLS3 with or without triparanol and desmosterol. Cells were pre-incubated with triparanol (2.5 μM or 5 μM) for 12 h, then incubated with desmosterol (40 μM) for 3 h. After drug washout, RSL3 (500 nM) was added for 4 h. **g** western blotting assay of DHCR24 in indicated HT1080 cells. **h** Cell death measurement of HT1080 cells expressing sgRNA-ctrl or sgRNA-DHCR24. Cells were pre-treated with desmosterol (40 μM) for 3 h. After washout, RSL3 (500 nM) was added for 4 h. **i** Cell death measurement of HT1080 cells treated with indicated drugs. Cells were pre-incubated with AY9944 (10 μM) for 12 h, then incubated with cholesterol (40 μM) for 3 h. After washing, RSL3 (500 nM) was added for 4 h. **j** Western blotting assay of DHCR7 in indicated HT1080 cells. **k, l** Cell death measurement of HT1080 cells expressing sgRNA-ctrl or sgRNA-DHCR7. Cells were pre-treated with cholesterol (40 μM) (**k**) or desmosterol (40 μM) for 3 h, followed by washout, then cells were treated with RSL3 (500 nM) for 4 h. Dead cells were stained with SYTOX Green (**a–c, f, h–i, k, l**). Data and error bars are mean ± SD, n = 3 (**a–l**) independent repeats. Significance in (**a–e**) was calculated using two-tailed unpaired Student's t-test. Significance in (**f, h**) was calculated using a one-way ANOVA with Tukey's post hoc test. Significance in (**i, k, l**) was calculated using a two-way ANOVA with Tukey's post hoc test.

**Both CoQ and squalene are required for cholesterol and desmosterol to suppress ferroptosis**. We next extended our studies on how perturbation of cholesterol metabolism modulates cellular CoQ levels. In an initial investigation, CH&Desmo did not influence mRNA levels of genes involved in CoQ10 synthesis downstream of farnesyl pyrophosphate (Fig. 3l–o, Supplementary Fig. 3n, o). Farnesyl pyrophosphate is a starting material for CoQ synthesis downstream of the mevalonate pathway, which is feedback inhibited by cholesterol. This inspired us to hypothesize that inhibition of cholesterol biosynthesis branch of the mevalonate pathway may result in a concomitant shift of the metabolic flux to CoQ10 synthesis. To test this possibility, we first examined expression of proteins involved in cholesterol biosynthesis upon CH&Desmo treatment. Indeed, cholesterol time-dependently decreased SQLE expression. A slight reduction was already detected at 1 h and became very obvious after 2 h treatment (Fig. 4a), which happened before mRNA level was downregulated (Fig. 4b), suggesting a posttranscriptional regulation. Further examination revealed a minimal effect (HMGCS1, HMGCR and FDPS) or a later response (LDLR) on other cholesterologenic proteins (Fig. 4a). Additionally, we showed that cholesterol decreased SQLE protein stability (Fig. 4c), which was completely rescued by MG132 but not 3-methyladenine (3MA), NH4Cl or chloroquine (CQ) (Fig. 4d), indicating a proteasomal mediated protein degradation. Similar results were also obtained with desmosterol (Supplementary Fig. 4a–c). Thus, CH&Desmo accelerate proteasomal degradation of SQLE protein to block cholesterol biosynthesis, which probably altered carbon flux to CoQ biosynthesis.

We next evaluated the role of SQLE in CH&Desmo mediated ferroptosis protection. Two independent gRNAs were used to generate SQLE null cell line by CRISPR-CAS9 (Fig. 4e). Ablation of SQLE strongly increased resistance to ferroptosis induced by RSL3, to a similar extent with CH&Desmo. More importantly, when these cells were treated with cholesterol or desmosterol, no further alteration of RSL3 sensitivity was observed (Fig. 4f, Supplementary Fig. 4d, e), suggesting a dominant contribution of SQLE degradation in the action of CH&Desmo.

The partial dependency on CoQ (see Fig. 3) but almost complete reliance on SQLE degradation suggests that a second factor is involved for the protection of CH&Desmo against ferroptosis. Careful literature search found a critical role of squalene in defending lipid peroxidation induced cell death[23]. To assay whether squalene also played a role, cells were preincubated with YM-53601, a chemical FDFT1 inhibitor. This showed apparent rescue of RSL3 induced ferropotic cell death (Supplementary Fig. 4g). However, genetic knockdown with siRNA or knockout by CRISPR gene editing displayed no change of

susceptibility to RSL3 treatment (Fig. 4g–j), and YM-53601 still inhibited ferroptosis in FDFT1 null cells (Supplementary Fig. 4h). These data suggested off-target effect of YM-53601, and more importantly, probable equal contributions of CoQ and squalene to ferroptosis resistance. We also tested another FDFT1 inhibitor TAK-475. As expected, supplementation of 5 μM TAK-475 for 12 h increased transcription of cholesterol biosynthetic genes in HT1080 cells (Supplementary Fig. 4i), and the effect of cholesterol or SQLE knockout on ferroptosis resistance was completely blocked (Fig. 4k, l). Moreover, whereas inhibition of CoQ synthesis partially restored CH&Desmo induced cell death resistance, as shown in Fig. 3c–f, combination of FDFT1 knockout with 4-CBA treatment or COQ2 knockdown completely blocked it (Fig. 4m, n, Supplementary Fig. 4j, k). To strengthen our conclusion, we generated FDFT1 and SQLE double knockout HT1080 cells. While SQLE knockout cells were resistant, the double knockout cells were similarly sensitive to ferroptosis with WT and FDFT1 KO cells (Supplementary Fig. 4l, m). In another attempt to expand our mechanistic studies, cells with ectopic expression of either FDFT1 or SQLE alone, or in combination were tested for their sensitivities to RSL3 (Fig. 4o). In the absence of cholesterol or desmosterol, overexpression of both proteins, but not either alone, significantly increased ferroptosis sensitivity (Fig. 4p, Supplementary Fig. 4n). As expected, CH&Desmo inhibited RSL3 induced cell death, and SQLE overexpression largely restored ferroptosis protection, which was further strengthened by FDFT1 overexpression, comparable to WT cell level (Fig. 4p, Supplementary Fig. 4n). These data together pinpoint essential roles of both squalene and CoQ as effective downstream factors for CH&Desmo in preventing ferroptosis.

**Cholesterol inhibits ferroptosis in doxorubicin and ischemia-reperfusion induced liver injury in mouse**. Ischemia-reperfusion injury (IRI) is a major challenge during liver surgical processes. Accumulating evidence suggests the involvement of ferroptosis in this pathogenesis[28]. Therefore, we were curious whether our cellular discoveries can be recapitulated in this in vivo model. To examine whether cholesterol protects mouse liver from ischemia-reperfusion induced damage, mice were fed chow diet or 0.1% cholesterol containing diet (CH diet). As shown in supplementary Fig 5a, b, serum and hepatic cholesterol levels were significantly increased after 12 h feeding. Immunoblot analysis indicated that, after CH diet feeding for 12 h, SQLE protein levels were much more reduced as compared to other proteins in the mevalonate pathway (Supplementary Fig. 5c). To maximize the effect of cholesterol, mice were fed chow diet or CH diet for 24 h. After exposure to non-lethal ischemia for 1 h, these mice were allowed to recover for another 24 h before sacrifice (Supplementary

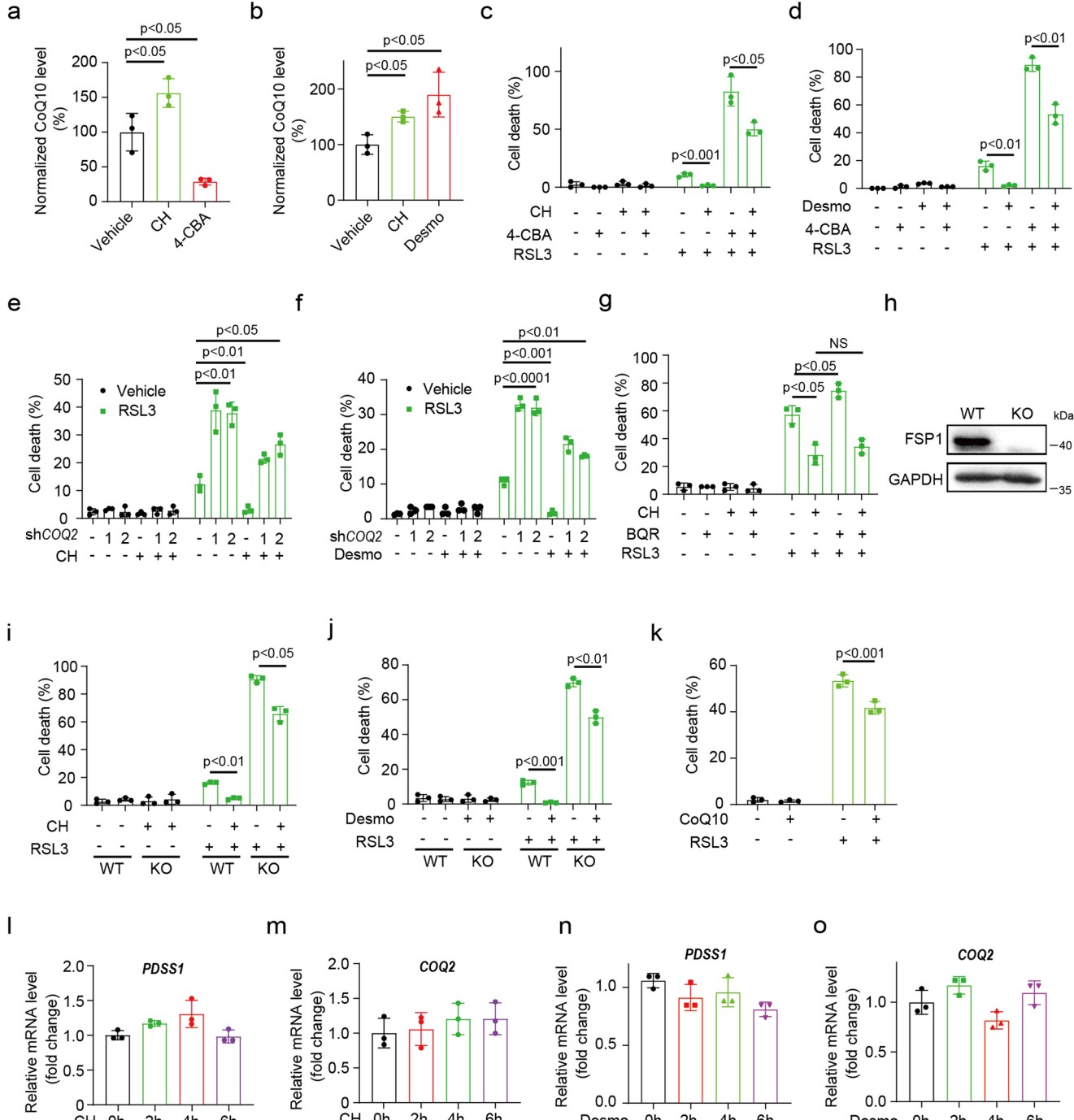

**Fig. 3 Cholesterol and desmosterol promote ferroptosis resistance partly through FSP1-CoQ10 axis. a**, **b** Mass spectrometric analysis of CoQ10 levels of HT1080 cells treated with cholesterol (40 μM), desmosterol (40 μM) for 6 h, or 4-CBA (5 mM) for 12 h. **c**, **d** Cell death measurement of HT1080 cells incubated with RSL3 with or without 4-CBA and cholesterol or desmosterol. Cells were pre-incubated with 4-CBA (5 mM) for 12 h, then co-incubated with cholesterol (40 μM) (**c**) or desmosterol (40 μM) (**d**) for 3 h, after which drugs were washed out and continued to be co-incubated with 4-CBA (5 mM) and RSL3 (500 nM) for 4 h. Uridine (200 μM) was continuously supplemented during the experiment. **e**, **f** Cell death measurement of wildtype and sh*COQ2* HT1080 cells in the presence of RLS3. Cells were pre-treated with cholesterol (40 μM) (**e**) or desmosterol (40 μM) (**f**) for 3 h, followed by thorough washout, and then treated with RSL3 (500 nM) for 4 h. Uridine (200 μM) was continuously supplemented during the experiment. **g** Cell death measurement of HT1080 cells pre-incubated with BQR (50 μM) for 12 h, then co-incubated with cholesterol (40 μM) for 3 h. RSL3 (500 nM) was added for 4 h after drug washout. **h** western blotting for FSP1 protein in indicated HT1080 cells. **i**, **j** Cell death measurement of wildtype or FSP1 knockout HT1080 cells. Cells were pre-treated with cholesterol (40 μM) (**i**) or desmosterol (40 μM) (**j**) for 3 h, then treated with RSL3 (500 nM) for 4 h after drug washout. **k** Cell death measurement of HT1080 cells pre-incubated with CoQ10 (250 nM) for 12 h, then co-incubated with RSL3 (500 nM) for 4 h. **l**–**o** Quantification of mRNA levels of *PDSS1* and *COQ2* in HT1080 cells treated with cholesterol (40 μM) (**l**, **m**) or desmosterol (40 μM) (**n**, **o**) for 0 h, 2 h, 4 h or 6 h. Data and error bars are mean ± SD, n = 3 (**a**–**o**) independent repeats. Significance in (**k**) was calculated using two-tailed unpaired Student's t-test. Significance in (**f**, **g**, **i**–**j**, **l**–**o**) was calculated using a one-way ANOVA with Tukey's post hoc test.

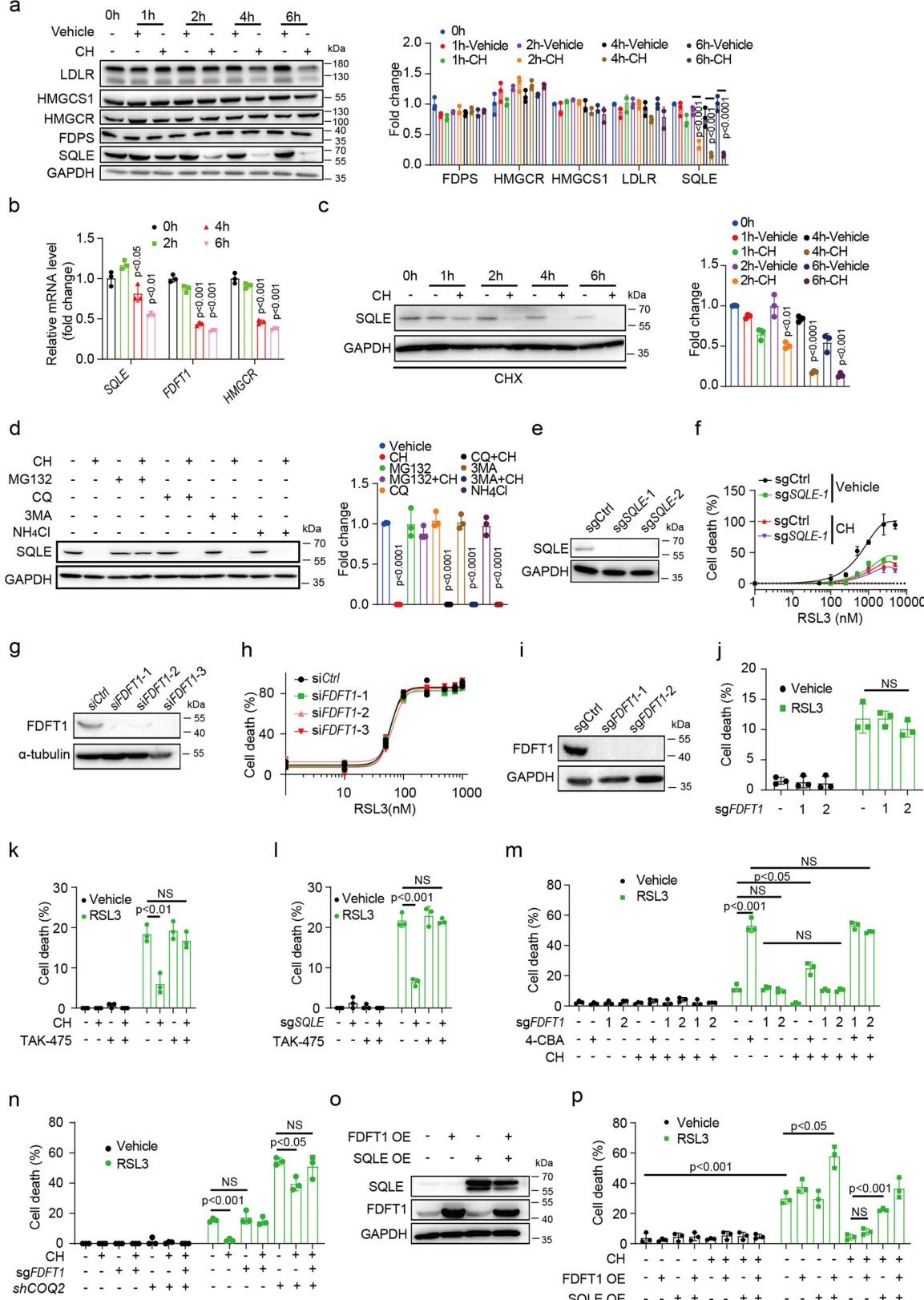

Fig. 5d). As shown in Fig. 5a, b, serum ALT and AST levels were significantly elevated after 24 h in chow fed mice, which was largely blocked after cholesterol feeding. *Ptgs2* is a key marker for ferroptosis[29]. Consistent with a known contribution of ferroptosis in IRI, mRNA levels of *Ptgs2* were much higher in ischemia group, which was prevented in CH diet fed mice (Fig. 5c). To

specifically address the functions of cholesterol in preventing oxidative stress induced by IRI, we examined the levels of 4-HNE and MDA, two markers of lipid peroxidation in vivo. Immunohistochemical staining showed that CH diet significantly diminished IRI induced increase of 4-HNE and MDA contents (Fig. 5d–f). As a positive control, Lipro-1 (Liproxstatin-1) largely

**Fig. 4 Both CoQ and squalene are required for cholesterol and desmosterol to suppress ferroptosis. a** Immunoblot analysis of indicated proteins in HT1080 cells after supplementation of cholesterol (40 μM) for indicated time. MβCD was used as vehicle control. **b** Quantification of mRNA levels of *SQLE*, *FDFT1*, *HMGCR* in HT1080 cells treated with cholesterol (40 μM) for 0 h, 2 h, 4 h or 6 h. **c** Immunoblot analysis of SQLE protein in HT1080 cells pre-treated with CHX (20 μg/ml) for 1 h, then co-treated with MβCD/cholesterol for indicated time. **d** Immunoblot analysis of SQLE protein in HT1080 cells with inhibitors pre-treated for 1 h followed by cholesterol (40 μM) for 4 h. Concentrations of inhibitors: MG132 (10 μM), CQ (10 μM), 3MA (5 mM), NH4Cl (5 mM). **e** Western blot analysis of knockout efficiency of SQLE protein in HT1080 cells. **f** Dose-dependent toxicity of RSL3 in HT1080 cells expressing sgRNA-Ctrl or sgRNA-*SQLE* with or without cholesterol (40 μM) treatment. **g** Western blot analysis of FDFT1 for HT1080 cells expressing siRNA-Ctrl or siRNA-*FDFT1*. **h** Dose-dependent toxicity of RSL3 in HT1080 cells expressing siRNA-Ctrl or siRNA-*FDFT1*. **i** Immunoblot assay of FDFT1 protein in wildtype and *FDFT1* knockout HT1080 cells. **j** Cell death measurement of HT1080 cells expressing sgRNA-Ctrl or sgRNA-*FDFT1* treated with RSL3 (500 nM) for 3 h. **k** Cell death measurement of HT1080 cells pre-incubated with TAK-475 (5 μM) for 12 h, then co-incubated with cholesterol (40 μM) for 3 h. RSL3 (500 nM) was added for 3 h after drug washout. **l** Cell death measurement of HT1080 cells expressing sgRNA-*SQLE*. Cells were pre-treated with TAK-475 (5 μM) for 12 h, then were incubated with RSL3 (500 nM) for 3 h. **m** Cell death measurement of HT1080 cells expressing sgRNA-Ctrl or sgRNA-*FDFT1*. Cells were pre-treated with 4-CBA (5 mM) for 12 h, then co-treated with cholesterol (40 μM) for 3 h. After drug washout, cells were incubated with RSL3 (500 nM) for 3 h. Uridine (200 μM) was continuously supplemented during the experiment. **n** Cell death measurement of HT1080 cells expressing sgRNA-*FDFT1* or shRNA-*COQ2*. Cells were treated with cholesterol (40 μM) for 3 h, followed by washout, then treated with RSL3 (500 nM) for 3 h. Uridine (200 μM) was continuously supplemented during the experiment. **o** Western blot of FDFT1 and SQLE protein in HT1080 cells after transducing indicated expression plasmids. **p** Cell death measurement of HT1080 in (**o**). Cells were treated with cholesterol (40 μM) for 3 h. After washout, cells were incubated with RSL3 (500 nM) for 3.5 h. Data and error bars are mean ± SD, *n* = 3 (**a**–**n**) independent repeats. Significance in (**a**, **c**, **d**, **h**), was calculated using two-tailed unpaired Student's t-test. Significance in (**b**, **j**–**n**) was calculated using a one-way ANOVA with Tukey's post hoc test. Significance in (**f**, **p**) was calculated using a two-way ANOVA with Tukey's post hoc test.

blocked IRI induced phenotypes. Mostly importantly, administration of TAK-475 largely reversed all phenotypes caused by cholesterol feeding (Fig. 5a–f), consistent with our in vitro data (Fig. 4k).

We further validated our findings in a different mouse model involving lipid peroxidation: doxorubicin induced liver damage (Supplementary Fig. 5e). Corroborating what we observed in the IRI model, CH diet significantly blocked elevated serum ALT and AST levels induced by doxorubicin injection (Fig. 5g, h), as well as markers of ferroptosis: mRNA levels of *Ptgs2* (Fig. 5i) and contents of 4-HNE and MDA in liver sections (Fig. 5j–l), which were again restored by TAK-475 (Fig. 5g–l). Together, our studies suggest physiological functions of cholesterol for protection against lipid peroxidation associated liver injury.

## Discussion

Alteration of metabolism has been increasingly linked to change of cell susceptibility to cell death. Multiple lines of evidence support this notion in the lipid peroxidation associated cell death, known as ferroptosis[16,19, 30]. Among various metabolic cues that regulate ferroptosis, lipid metabolism, in particular, fatty acid metabolism has been most intensively investigated. Incorporation of polyunsaturated fatty acids into phospholipids, the major component of cell membranes, represents a conserved vulnerability that necessitates complex antioxidative defense system. Although the role of phospholipid in regulation of ferroptosis is well known, surprisingly, the role of cholesterol, another critical component of cell membrane, and other sterol metabolites are far less studied.

In this work, we specifically target sterol metabolites for possible ferroptosis modulators. In all sterols we have initially tested, 7-DHC stands out to be the most effective anti-ferroptotic agent, which is followed by CH&Desmo, who display comparative activities. Consistent with our findings, a recent study identified *DHCR7* as pro-ferroptotic gene in a genomic wide CRISPR screen. Inactivation of DHCR7 induced 7-DHC accumulation and subsequent resistance to ferroptosis[14]. Our further mechanistic studies focusing on CH&Desmo lead to the conclusion that SQLE degradation mediated by these two sterols likely increases intracellular levels of both squalene and CoQ, two known metabolites, to combat ferroptosis (Fig. 6). This is supported by the following observations: first, both sterols specifically induce a proteasome dependent degradation SQLE but no other proteins

in the mevalonate pathway, as manifested by earlier decrease of SQLE protein than mRNA levels, and rescuing effect of proteasome but not autophagy inhibitors (Fig. 4). Second, although genetic deletion of *SQLE* or CH&Desmo supplementation enables cells more resistant to ferroptosis, treatment of *SQLE* knockout cells with these two metabolites exhibit no additive or synergistic effect, suggesting involvement of the same pathway. This is further supported by the evidence that ectopic expression of SQLE near completely restores CH&Desmo induced inhibition on ferroptosis. Finally, while inhibition of CoQ synthesis by 4-CBA or knockdown of *COQ2*, a key protein involved in CoQ synthesis only partially abrogates functions of CH&Desmo, combinatorial blockade of CoQ and squalene synthesis entirely abolishes the protecting effect of CH&Desmo on ferroptosis. Taken together, our data demonstrate important functions of CH&Desmo in inhibiting ferroptotic cell death.

Sterols are a group of lipids composed of four rigid hydrocarbon rings fused to a flexible hydrocarbon tail on one side and a hydroxyl group on the opposite side. Modifications on the ring structure or/and the tail give rise to molecules with distinct properties. For example, both lanosterol and 25-hydroxycholesterol (25HC) stimulate degradation of HMG CoA reductase (HMGCR), whereas even 10 times more cholesterol has no effect. In contrast, 25HC, but not lanosterol binds and induces conformational change of SCAP to inhibit SERBP processing[31,32]. In our screen, 7-DHC manifests highest activity against ferroptosis as compared to CH&Desmo. Interestingly, washout experiment indicates that 7-DHC also sustains inhibitory effect on ferroptosis longer than CH&Desmo (Fig. 1). These data suggest a possible different mechanism of 7-DHC with CH&Desmo, but similar mode of action for the latter two sterols. In line with this, fluorescence-enabled inhibited autoxidation (FENIX) assay indicates that, unlike 7-DHC, CH&Desmo have no effect on the rate of phospholipid oxidation ([14] and Supplementary Fig. 3), and CH&Desmo induce SQLE degradation with similar kinetics (Fig. 4). Interestingly, studies have demonstrated that mice lacking cholesterol but accumulating desmosterol by targeted disruption *DHCR24* have only mild phenotype[33], owing to the functional similarities between cholesterol and desmosterol[34, 35]. Given our screen mainly focuses on intermediates of the mevalonate pathway, whether there are other sterols/steroids that can counter ferroptotic cell death warrants further investigation.

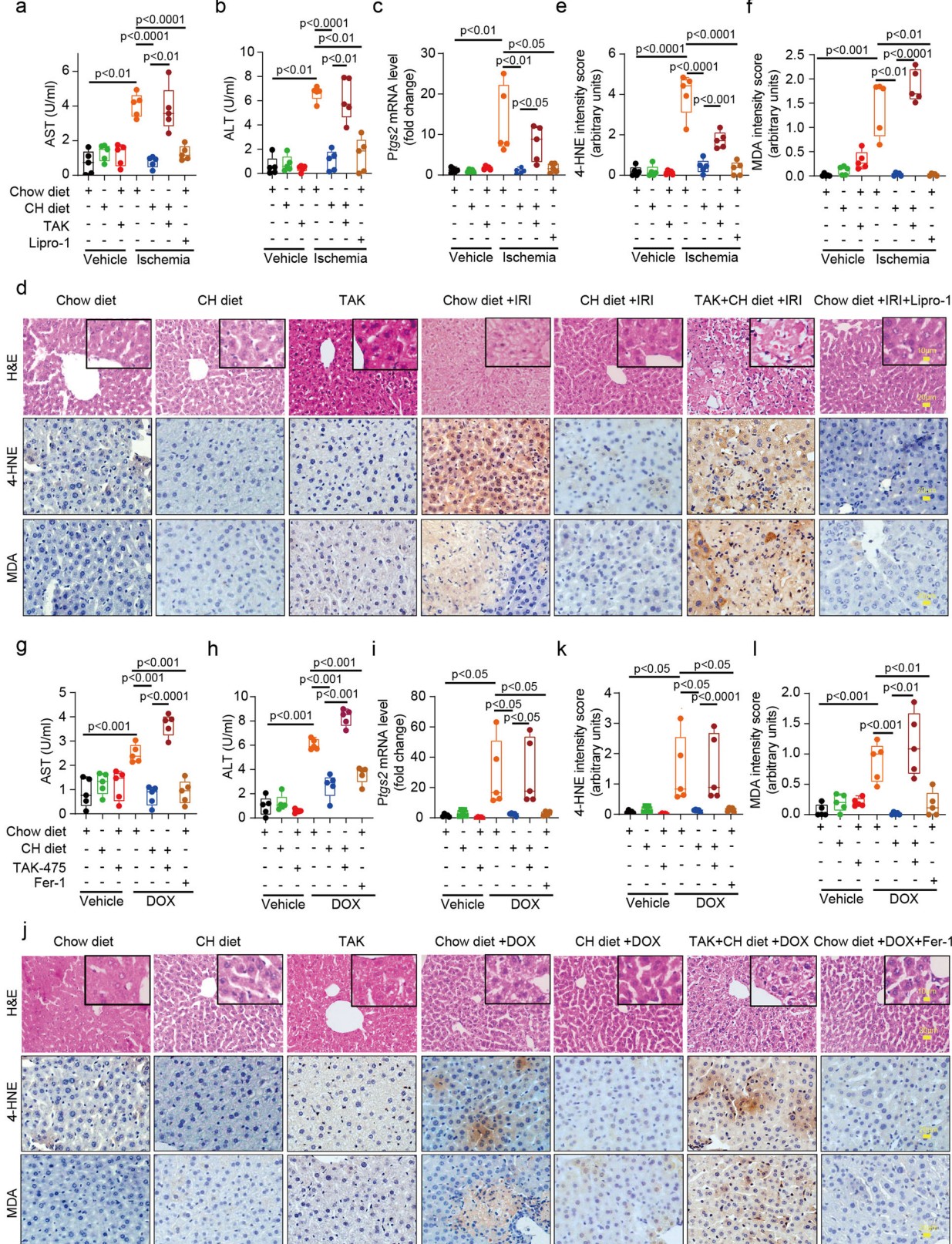

**Fig. 5 Cholesterol inhibits ferroptosis in doxorubicin and ischemia- reperfusion induced liver injury in mouse. a**, **b**, **g**, **h** Analysis of serum AST (**a**, **g**) and ALT (**b**, **h**) levels in mice from indicated groups. **c**, **i** Relative mRNA expression of *Ptgs2* in livers of indicated groups. **d**–**f**, **j**–**l** Representative immunohistochemical (IHC) images and statistical analysis of 4-HNE and MDA staining from mice liver tissues in indicated groups (scale bar, 20µm). Data and error bars are mean ± SD, *n* = 5/group. Significance in (**a**–**f**, **g**–**l**) was calculated using two-way ANOVA with Tukey's post hoc test.

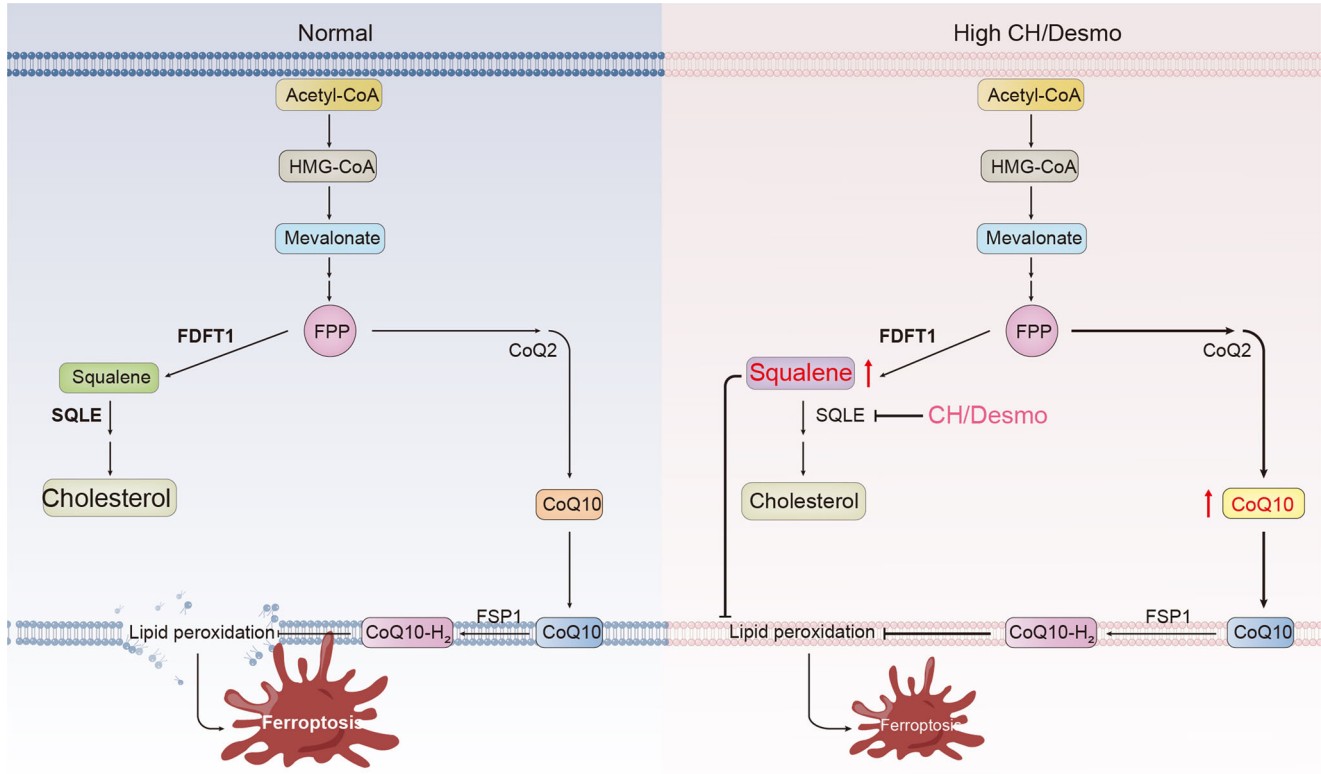

**Fig. 6 Schematic diagram of the mechanism of action of CH&Desmo.** Under normal conditions, cells maintain basic levels of cellular squalene and CoQ10 to combat low extent of lipid peroxidation and ferroptosis. The balance can be easily disrupted by various stimuli to induce ferroptotic cell death. Under circumstances where cells are loaded with excess CH&Desmo, SQLE degradation is accelerated, which blocks carbon flux through squalene to cholesterol. Consequently, cellular levels of both squalene and CoQ10 are increased, and ferroptosis is inhibited.

CoQ has been long known to be capable of protecting lipid from peroxidation[36, 37]. However, it was not until recently that a specific role of this metabolite in inhibition of ferroptosis was documented[21, 22]. On the other hand, squalene protects skin from UV irradiation induced lipid peroxidation[38], and loss of SQLE activity in ALK+ anaplastic large cell lymphoma (ALCL) cells results in accumulation of squalene and protects these cells from ferroptosis[23]. In spite of these, the mechanism by which cells regulates levels of intracellular CoQ and squalene as response to metabolic stress has not been characterized. Given that the mevalonate pathway is responsible for the production of both metabolites, and this pathway is subject to intricate feedback regulations by sterols, it is not unreasonable to hypothesize that cholesterol metabolism could be one type of such metabolic stress. Indeed, cholesterol and desmosterol, but not 7-DHC, were able to promote SQLE degradation to increase squalene levels[9]. Our data confirms and further expands this conclusion by suggesting a further increase of CoQ levels besides that of squalene (Fig. 4). This enhanced CoQ levels is likely caused by a metabolic flux shift to non-sterol branch as a consequence of SQLE loss of function.

As an initial effort to characterize how metabolic flux shift between CoQ and squalene affects the progression of ferroptosis, YM-53601 was used to inhibit FDFT1, the enzyme located at the branch point in the mevalonate pathway. Pre-treatment of HT1080 cells with this compound markedly increased resistance to RSL3 induced cell death (Supplementary Fig. 4), consistent with what has been observed by Shimada et al. [39]. However, neither siRNA mediated knockdown nor knockout of *FDFT1* through CRISPR altered cell viability in the presence of RSL3, suggesting an off-target effect of this drug. Thus, reduced levels of squalene by blocking FDFT1 activity is likely complemented by

more CoQ biosynthesis, with a neat "zero" effect, at least in the context of ferroptosis induced by RSL3.

Besides participating in the feedback regulation of cholesterol biosynthesis by promoting SQLE degradation, cholesterol is known to have many other functions, such as control of membrane rigidity and signaling nutrient availability through SLC38A9-mTORC axis. Therefore, we can't exclude the possibility that other pathways may contribute to ferroptosis inhibitory effect of cholesterol. As our manuscript was being prepared, two laboratories have reported anti-ferroptotic functions of cholesterol in different cells. In one report, Liu et al. found a protective role of cholesterol in maintenance and myeloid bias of long-term hematopoietic stem cells (LT-HSCs) through defending against ferroptosis[40]. It was proposed that upregulation of SLC7A11/GPX4 expression by SLC38A9-mTORC signaling axis played a central role downstream cholesterol. However, we were not able to detect this change in HT1080 cells with either 40 µM cholesterol or desmosterol treatment up to 6 h, which is much longer than needed for these two sterols to take effect (3 h pretreatment plus 2 h RLS3 treatment in lipid peroxidation assay, Fig. 2). Additionally, the effects of CH&Desmo seem to be independent of SLC7A11 or GPX4 (Fig. 1). The discrepancy between our results and those of Liu et al. could be explained by different dose or/and time period of cholesterol treatment. Alternatively, different cells (LSCs and HT1080 cells in this case) may have distinct responses to the same stimulation, as has been shown by others[41]. In another work, cholesterol was shown to decrease membrane fluidity and promote lipid raft formation, thereby restraining lipid peroxidation in cancer cells[42]. Interestingly, we also observed SQLE independent effect in HT1080 cells, where cells were supplemented with 50 µM instead of 40 µM cholesterol (Supplementary Fig. 4f), suggesting other pathways taking effect at high

cholesterol concentration. Together, this suggests that the functions of cholesterol in ferroptosis resistance could be multifaceted.

In summary, our work has uncovered a perspective of cholesterol metabolism in protecting cells from ferroptosis. The benefits of low dosage cholesterol uptake on perfusion-ischemia and doxorubicin induced hepatotoxicity, as exemplified in our study, may pave a way to therapeutic strategies that alleviate lipid peroxidation associated tissue damage. Our study also provides conceptual advance whereby targeting SQLE instead of FDFT1 for degradation by sterols in a feedback loop regulation of cholesterol biosynthesis may further provide evolutionary advantages for mammalian cells to evade oxidative stress induced cell death. It should be noted that low-dosage and short-time cholesterol feeding strategy was adopted in our mouse studies. Since chronic and/or overaccumulation of cholesterol are known to be cytotoxic, care should be taken for the future translational applications of current research.

## Methods

**Cell culture**. HT1080 and 786-O cell lines were purchased from the Cell Bank of the Chinese Academy of Science (Shanghai, China) and have been tested to confirm no mycoplasma contamination. All cell lines were maintained at 37 °C and 5%$CO_2$. The cells were cultured in Dulbecco's modified Eagle's medium (DMEM) medium (Gibco), supplemented with 10% fetal bovine serum (Biological Industries, Israel) and 1% Penicillin/Streptomycin (Gibco) solution. HT1080 $GPX4^{-/-}$ cell line was a gift from Dr. Minghui Gao (Harbin Institute of Technology, China).

**Plasmids construction and cell transduction**. The gene overexpression plasmids were constructed by standard molecular cloning: full length SQLE was cloned into pLX304 and full length FDFT1 was cloned into pLenti vectors; For shRNA-mediated COQ2 gene knockdown, oligonucleotides with the following sequences (5'-CCATTATAATTCTCAC CTGAA-3'; 5'-CCTGAGGATTGTTGGAATAAA-3') were cloned into pLKO.1 vector. Plasmids were transfected into HEK293T cells together with lentiviral packaging vectors Delta-8.9 and VSV-G using PEI transfection reagent (Polysciences, 23966). For virus infection, HT1080 cells were spin infected in 12-well tissue culture plates using 8 μg/ml of polybrene at 1200 g for 60 min and selected by puromycin (0.5 μg/ml) or blasticidin (4 μg/ml) for 3 days. The expression or knockdown efficiency was verified by immunoblotting or RT-qPCR analysis. FDFT1 siRNA was purchased from Shanghai GenePharma Co., Ltd. siRNAs were transduced to HT1080 cells with lipofectamine3000 (Sigma) according to the manufacturer's protocol. After 48 h of transfection, some cells were seeded into 96-well plates for experiments, and the rest cells were collected for immunoblotting to detect knockdown efficiency. The sgRNA sequences were designed from the website (http://crispor.tefor.net/). These sgRNAs were cloned into plenti-CRISPR-V2 vectors.

sgSQLE: 5'-CCATCTGCAACAACAGTCAG-3', 5'-CCACTGACAATTCTCATCTG-3'; sgFDFT1: 5'-CAGTTTCGCAGCTGTTATCC-3', 5'-CTCTCCATGAACCGCCAGTC-3'; sgDHCR7: 5'-CCACAAGGTATAGAGCTGGGCGG-3', 5'-TGCGAAGGACAGGTTGATGAGGG-3';

sgDHCR24: 5'-AGTGCACCCCAAAGGAAATG-3', 5'-AAGTTTGTCCGCAGCGTGCA-3'; sgGPX4: 5'-GGCGCTAGCTCACCATGGTCC-3';

sgFSP1: 5'-GCACTCTCATTCACTCCCAAG-3';

Plasmid were transfected into HEK293T cells with lentiviral packaging vectors psPAX2 and pMD2.G using PEI transfection reagent (Polysciences, 23966). After transduction and selection

using puromycin, colonies grown from single cell were picked and expanded for experiments.

**Chemicals and antibodies**. Erastin (S7242), RSL3 (S8155), ferrostatin-1 (S7243), liproxstatin-1 (S7699), Lanosterol (S4755), Avasimible (S2187), Chloroquine (S4157) were purchased from Selleck. SYTOX Green (S7020) and BODIPY-C11 581/591 (D3861) were purchased from Invitrogen. 27-Hydroxycholesterol (HY-N2371), 24,25-Dihydrolanosterol (HY-W040264), Zymosterol (HY-114297), Lathosterol (HY-113486), desmosterol (HY-113224), 7-Dehydrocholesterol (HY-113279), Triparanol (HY-W019996), U18666A (HY-107433), YM53601(HY-100313A), MG132(M8699), 3-Methyladenine (3-MA, M9281), Ammonium chloride (NH4CL, HY-Y1269), Cycloheximide (HY-12320) were purchased from MCE. 25-Hydroxycholesterol (H1015), cholesterol (C3045), AY9944 (190080), MβCD (332615), 4-CBA (135585), Uridine (U3003), methylcellulose (M0512) were purchased from Sigma. TAK-475 (CM06516) was from proteintech. BQR (T4220) was purchased from Topscience.

Antibodies for SQLE (12544-1-AP), DHCR24 (10471-1-AP), NQO1 (11451-1-AP), α-Tubulin (11224-1-AP), GAPDH (6004-1-AP), FSP1 (20886-1-AP), DHODH (14877-1-AP), GCH1 (28501-1-AP), Vinculin (66305-1) and β-actin (20536) were purchased from Proteintech. FDFT1 (A4651), LDLR (A14996), HMGCS1 (A3916), FDPS (A5744) antibodies were purchased from ABclonal. HMGCR (ab174830), GCLM (ab126704), GPX4 (ab125066), ACSL4 (ab155282), 4-HNE (ab46545), MDA (ab243066) antibodies were obtained from abcam. SLC7A11 (#12691) and HO-1 (#70081) antibodies were from CST.

**Solubilization of sterols**. Sterol-MβCD complexes were prepared according to the method described in ref. [43] with minor modifications. In brief, 50 mg/mL sterol stocks in 1:1 chloroform: methanol were prepared and evaporated under a stream of nitrogen flow. Dried sterols were mixed with 38 mM MβCD solution prepared in Opti-MEM medium so that sterols: MβCD ratio were 1:10. Sterol suspensions were incubated on a shaker O/N at 4 °C until sterols were dissolved. Sterol crystals were subsequently removed via 0.45 μm filtration.

**Cell viability assay**. For cell viability assay, $5 \times 10^3$ cells per well were seeded in 96-well plates. 12 h later, cells were treated with indicated concentrations of RSL3 with or without other drugs. Cells were then incubated with 10 μl CCK-8 solution for 1 h at 37 °C, 5%$CO_2$. The spectrophotometric absorbance of the samples was read using a ELISA microplate reader at 450 nm with 630 nm as a reference wavelength. Percentage of cell viability was then calculated after subtracting background absorbance of cell culture medium[44].

**Cell death assay**. For cell death assay, $5 \times 10^3$ cells per well were seeded in 96-well plates and cultured overnight. After drug treatment for indicated time, 500 nM final concentration of SYTOX Green Nucleic Acid dye was added into each well, cells were further incubated at 37 °C for 15 min, and images of at least three randomly chosen fields were captured by an inverted fluorescence microscope (Olympus, Japan). Cells with green fluorescence were considered as dead cells, and cell death rate was eventually calculated by the ratio of dead cells/ (living cells + dead cells)[19].

**Immunofluorescence of BODIPY-C11 581/591staining**. HT1080 cells were seeded on round slides in 24-well plates. After O/N culture, cells were pretreated with intermediates of cholesterol biosynthetic pathway for 3 h, drugs were washed out with

serum-free medium, followed by incubation with 5 μM BODI1PY-C11 581/591 and RSL3 (250 nM) at 37 °C for 2 h in the dark. Cells were then washed twice with PBS and fixed with 4% paraformaldehyde for 15 min at room temperature. Cells were then washed twice with PBS followed by one wash with ddH$_2$O. The slides were mounted on the cover glasses and observed under the Andor Dragonfly 200 confocal microscope. Fluorescent images were acquired with 488 nm and 561 nm excitation for oxidized and reduced forms of dye respectively[19].

**FACS-based lipid peroxidation assay.** $2 \times 10^5$ HT1080 cells were seeded per well in 12-well plates. After O/N culture, cells were pretreated with sterols for 3 h. After thorough washout, cells were treated with RSL3 (250 nM) for 2 h. Then cells were trypsinized and collected, washed once with PBS. Then each sample was added with 5 μM BODI1PY-C11 dye in PBS and incubated at 37 °C for 15 min in the dark. Cells were then washed twice with PBS followed by re-suspending in 500 μl PBS. The levels of lipid ROS were analyzed through FL1/FL2 chanel using a Becton Dickinson FACS Calibur machine, and quantitation of lipid peroxidation was achieved by normalizing FL1 to FL2 channel in FlowJo V10 software. Flow cytometry gating strategy was described in Supplementry Methods.10,000 cells were analyzed for each sample[45].

**Filipin staining.** Cells were seeded at a density of $5 \times 10^5$ cells per well in 6-well plates. The next day, cells were pretreated with 40 μM sterol intermediates (CH/ Desmo/7-DHC/ Latho/ Zymo/ Lano) for 3 h. Then cells were trypsinized and collected, washed once with PBS. Cells were then fixed with 4% paraformaldehyde for 15 min at room temperature. 300,000 cells for each sample were used for filipin staining (50 μg/mL in PBS) for 1 h at room temperature[46]. Cells were then washed twice with PBS followed by re-suspending in 500 μl PBS. Intracellular cholesterol levels were analyzed through PB450 channel using Beckman cytoflex machine, and data were analyzed by CytExpert 2.4. 10,000 cells were analyzed for each sample. Flow cytometry gating strategy was described in Supplementry Methods.

**FENIX assay.** Egg PC liposomes (extruded to 100 nm, 1 mM), STY-BODIPY (1 μM), and cholesterol (40 μM), desmosterol (40 μM), 7-DHC (40 μM), or vehicle (MβCD) were vortexed in PBS (10 mM, 150 mM NaCl, pH7.4), then 200 μl aliquots were incubated in a black 96-well plates (polypropylene, Nunc) for 20 min at 37 °C. DTUN (200 mM in EtOH) was added to the aliquots. The plate was mixed for 5 min and kinetic data of STY-BODIPY$_{OX}$ was acquired at 488 nm excitation and 518 nm emission by Mithras LB940 microplate reader (Berthold Technologies)[29].

**Western blotting.** Cell or tissue samples were lysed with RIPA buffer containing 1% protease inhibitor, and the protein concentration of cell lysates was measured by BCA protein assay kit (Beyotime, China). Total proteins were separated by 10% SDS-PAGE, then transferred onto nitrocellulose membrane (Millipore Corp, Billerica, MA, USA). The membrane was blocked with 5% milk-TBST at room temperature for 1 h, primary antibodies were incubated overnight at 4 °C, followed by affinipure peroxidase goat anti-rabbit or mouse IgG(H + L) secondary antibodies for 1 h at room temperature. Signals were detected by enhanced chemiluminescence solution (GE Healthcare) for visualizing protein expression.

**Extraction of CoQ10.** $10^6$ cells were collected, washed once with PBS, and centrifuged at 13,000 rpm for 3 min. After discarding

PBS, 100 μl of MilliQ water was added to cell pellets to break the cells. Aliquots of samples were taken to measure the protein concentration. Then 500 μl of hexane and 400 μl of methanol were added to the remaining samples, with 0.75 μM COQ9 supplemented as internal control. Samples were sonicated twice, vortexed and centrifuged at 13,000 rpm for 5 min. The upper phase liquid was transferred to a new EP tube after delamination, and 500 μl of hexane was added to the lower phase for a second extraction. Upper phases of the two extractions were mixed and dried with a stream of N$_2$ until a transparent gel-like precipitate was obtained. The dried samples were re-dissolved with 150 μl hexane and applied to extraction column (Strata SI-1 Silica (55 μm, 70 A) pre-balanced with hexane. Columns were washed with 750 μl hexane followed by 750 μl hexane/acetic acid ethyl ester (18/1, v/v). After wash, 750 μl hexane/acetic acid ethyl ester (9/1, v/v) was used to elute bond CoQ, and the eluents were collected and dried with N$_2$. Samples were sent to Metabonomics Platform for LC-MS analysis at Tsinghua University[21].

**Mice and treatment.** 8-week old male C57BL/6 J mice were purchased from Charles River Laboratories (Beijing, China). Mice were fed customized control chow diet containing 0.5% bile salt and 10% fat, or high cholesterol diet (CH diet, chow diet supplemented with 0.1% purified cholesterol) (Trophic, Nanjing, China, TP23302). All animal studies were performed in accordance with the Guidelines for the Care and Use of Laboratory Animals and were approved by the Institutional Animal Care and Use Committees at Shandong University. Animals were euthanized using CO$_2$.

**In vivo drug treatment-Doxorubicin.** All animals were acclimatized for 1 week before experimentation. Mice were divided into seven groups (Chow diet, CH diet, TAK, Chow diet +DOX, CH diet +DOX, TAK + CH diet +DOX and Chow diet +DOX +Fer-1), with 5 mice in each group. Mice were treated with TAK-475 (50 mg/kg) by oral gavage in a 0.5% methylcellulose suspension vehicle solution. Then all mice were fed customized diets for 24 h. Subsequently, 20 mg/kg doxorubicin or the same volume of vehicle (saline) was given by a single intraperitoneal injection. For the Chow diet + DOX +Fer-1 group, mice were injected intraperitoneally with 10 mg/kg ferrostatin-1 24 h before doxorubicin administration. Blood and fresh liver tissues were harvested 24 h after doxorubicin injection.

**Hepatic ischemia/reperfusion injury model.** All animals were acclimatized for 1 week before experimentation and divided into seven groups (Chow diet, CH diet, TAK, Chow diet +IRI, CH diet +IRI, TAK + CH diet +IRI and Chow diet +IRI +Lipro-1) with 5 mice in each group. Mice were treated with TAK-475 (50 mg/kg) by oral gavage in a 0.5% methylcellulose suspension vehicle solution. Then all the mice were pretreated with customized diet for 24 h. For ischemia/reperfusion surgery, mice were randomly distributed and anesthetized by intraperitoneal injection of sodium pentobarbital (30 mg/kg, Sigma). Then they were fixed on a mouse plate maintained at a constant temperature of 37 °C. After the mice no longer responded to the pain stimuli, an incision was made along the midline of the abdomen. Then, the structure of the hepatic portal was carefully separated, and the blood vessels in the left and middle lobes of the liver were blocked with sterile arteriole clips, which instantly caused partial hepatic ischemia. The abdomen was covered with normal saline gauze. After 60 min, the arteriole clip was taken out to re-establish blood flow and reperfusion. The abdominal incision was sutured, and mice were heated on an electric blanket until they woke up from anesthesia. After surgery, mice were returned to the rearing cage

for continued feeding for 24 h until being euthanized. For the Chow diet +IRI +Lipro-1 group, mice were injected intraperitoneally with 10 mg/kg Liproxstatin-1 one hour before IRI surgery. Mouse blood and fresh liver tissues were harvested 24 h after IRI surgery for further studies.

**Immunohistochemistry**. Liver tissues of mice were isolated, fixed in 4% paraformaldehyde for 48 h. The fixed tissues were dehydrated, embedded and paraffin sectioned. Immunohistochemical staining was carried out with anti-4-HNE (1: 200) and anti-MDA (1: 200) antibodies. In brief, the slices were first dewaxed in xylene, then hydrated with serially diluted alcohol, followed by incubation with 2% Tween-20 for 20 min. Antigen was repaired by treating slices with citrate buffer (pH 6.0) for 15 min in a water bath at 98 °C. After returning to room temperature, slices were incubated with 3% $H_2O_2$ to reduce the activity of endogenous peroxidase. 5% BSA was used to block nonspecific epitopes and then primary antibodies were incubated at 4 °C overnight. Subsequently, slices were incubated with biotin-labeled secondary antibody and SABC reagent for 30 min, according to the manufacturer's instructions. 3,3-diaminobenzidine and H&E staining were used for detection of antigens and nucleus respectively. Immunohistochemical images were taken on a multispectral scanning microscope imaging system (TissueGnostics, Austria), and the average optical density (MOD) was calculated by the software Image Pro-plus.

**Cholesterol content determination**. Cholesterol content of HT1080 cells was measured with Amplex Red Cholesterol Assay Kit (Molecular Probes, A12216). RIPA Buffer was used to lyse the cells, and an aliquot of cell lysates was taken to determine the protein concentration by BCA method (Beyotime, P0011). For quantification of cellular cholesterol content, 5 µl cell lysate or cholesterol standard solution was incubated in a 96-well plate with reagents provided in the kit, and the fluorescence was read by Envision (PerkinElmer, 1905116 S). For measurement of cholesterol levels of liver tissues, the lysis solution from the Tissue TC assay kit (APPLYGEN, E1015) and steel balls were added to the liver tissues. Samples were grinded (60 Hz, 3 min). Aliquots of cleared cell lysates were taken to measure the protein concentration. Cholesterol content was assayed in a 96-well plate according to the instructions. For mouse serum samples, 5 µl serum or standards were used. Cholesterol was detected with the kit purchased from APPLYGEN (E1005).

**Measurement of serum AST and ALT levels**. Serum AST and ALT levels were measured using the Micro Glutamic-oxalacetic Transaminase (GOT) Assay Kit (Solarbio, BC1565) and Glutamic-pyruvic Transaminase (GPT) Activity Assay Kit (Solarbio, BC1555) in accordance with the manufacturer's instructions.

**Statistics and reproducibility**. All statistical analyses were performed using the GraphPad Prism 8.0 and Microsoft Excel. All data are presented as mean ± SD except where indicated. All comparisons were tested using unpaired two-tailed Student's t-test or one/two-way ANOVA with Tukey's post hoc test, with a confidence interval (CI) of 95%. $P \leq 0.05$ was considered statistically significant. Blinding was performed for in vitro experiments with data analysis by different operators.

## Data availability

All data supporting the findings of this study are available within the paper and/or the Supplementary Information. Uncropped and unedited blot/gel images are available in

Supplementary Information. The source data for the graphs and charts is available as Supplementary Data 1 and any remaining information can be obtained from the corresponding author upon reasonable request.

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

## Acknowledgements

We thank Dr. Minghui Gao (Harbin Institute of Technology, China) for general gift of GPX4$^{-/-}$ cells. We also thank Translational Medicine Core Facility and Core facility of school of basic medical sciences of Shandong University for consultation and instrument availability that supported this work. This work was supported by grants from Shandong Natural Science Fund (21510005202107, J.L.), National Key R&D Program of China (2022YFA0912600, B.C.), National Natural Science Foundation of China (grant no. 32000515, B.C.), Natural Science Foundation of Shandong Province (ZR2020QC074, B.C.), Qilu Young Scholars Program (B.C. and J.L.), Multidisciplinary Research and Innovation Team of Young Scholars (21500061510007, J.L.).

## Author contributions

Conceptualization, J.L., and B.C.; Investigation, Q.S., D.M.L., W.W.C., L.X.H., H.M.C., R.H.Z., S.L., and X.Z.; formal analysis, J.L.G., Y.L., J.L., B.C.; Writing-Original Draft, Q.S., D.M.L., W.W.C., J.L. and B.C.; Writing-Review & Editing, J.L.G., Y.L., J.L., and B.C.; Funding Acquisition, J.L. and B.C.; Supervision, J.L. and B.C.

## Competing interests

The authors declare no competing interests.
