## [Peer Review File · Communications Biology]

Referee expertise:

Referee #1: Cholesterol metabolism, ferroptosis

Referee #2: lipid metabolism, ferroptosis

Referee #3: lipid metabolism

Reviewers' comments:

Reviewer #1 (Remarks to the Author):

Sun et al demonstrates a causal link between cholesterol and ferroptosis susceptibility as well as its clinical relevance. Although the notion that mevalonate pathway regulates ferroptosis susceptibility has been well recognized, potential interests exist regarding the its interplay with cholesterol homeostasis and its implication in hepatotoxicity. Overall, this study is well designed, yet several aspects need to be better interpreted.

Major comments:

1. Figure 1a: The levels of lipid peroxidation should be provided to demonstrate that the cells were really protected from ferroptosis.
2. Figure 2d, e: Whether the background lipid increase was corrected by FL2 channel when detecting lipid peroxidation using C11 BODIPY staining?
3. Figure S2e, f: The legend should be carefully checked. Cysteine starvation induced a nearly 90% of cell death (Figures S1e, S1h, S2c), while this was not observed in Figure S2e, f.
4. Figure 2a-c: Why did RSL3 induce a discrete frequency of ferroptosis.
5. Figures 3, 4: CH&Desmo also significantly attenuated ferroptosis of CoQ10-depleted or FSP1-deleted cells. Does that mean CH&Desmo inhibit ferroptosis independently of FSP1-CoQ10 axis? In Figure 4, FDFT1 deletion alone was sufficient to abrogate the protective effect of CH on ferroptosis. The contribution of FSP1-CoQ10 axis remains doubtful.
6. Appropriate statistical analysis should be used, such as one/two-way ANOVA and multiple comparison for three or more groups.

Minor comments:

1. Some grammatical mistakes such as "that involve" in Abstract, "array of" in Introduction, "are able to", "independent on each other" in Results and etc. The manuscript should be thoroughly checked.
2. Supplementary Figures particularly Figures S1 and S5 should be better cited and arranged to easily follow the results.
3. Figures 1, 5: What are Fer-1 and Lipo?

Reviewer #2 (Remarks to the Author):

In this manuscript, Sun et al. suggest that cholesterol inhibits ferroptosis via CoQ10 and squalene metabolism and further show that cholesterol diet can protect from hepatotoxicity. Overall, the majority of experiments are well-designed, well-conducted, and clear enough to support the author's claim. There are only minor basic issues.

1. The data suggest that the levels cholesterol rather than its intermediates themselves eventually affect CoQ10 and squalene metabolism to inhibit ferroptosis. However, the intermediate metabolism seems to be focused in the manuscript and supplementary figures, so a graphic summary of ferroptosis inhibition by cholesterol would be helpful for readers.
2. Why several intermediates for cholesterol synthesis are dispensable for ferroptosis suppression? Did not they contribute to total cholesterol levels?

3. If desmosterol inhibits ferroptosis via cholesterol synthesis, the inhibitory effect of desmosterol should be tested in DHCR7 KO cells
4. In addition, as the authors suggest, the possible additional mechanism by which 7DHC inhibits ferroptosis via direct inhibition of autoxidation may be indicated in the figure. Furthermore, if possible, direct observation of 7DHC-mediated inhibition of autoxidation is required (Supp Fig. S3d, related to page 6).
5. The levels of cholesterol in cells treated with M β CD should be provided. Because the direct observation of altered cholesterol levels upon avasimible and M β CD is critical, please provide these data (supplementary fig. 2d) in the main figure.
6. The involvement of CoQ10 in CH-mediated ferroptosis suppression is still unclear. First, recycling of CoQ10 by FSP1 may be much more critical for ferroptosis suppression rather than the levels of CoQ10. Partial inhibition of ferroptosis by CH in FSP1 KO cells may suggest the requirement of FSP1 for CH-mediated ferroptosis suppression but it also imply that CH can also inhibit ferroptosis in the absence of FSP1. Could the author see the inhibition of ferroptosis by supplementing CoQ10 that can lead to 2 fold increase? This can be just discussed.
7. Fig. 5a, Lipo may be corrected to Lipo-1.

Reviewer #3 (Remarks to the Author):

This study is interesting. I went through the paper. Please consider the following comments to improve the quality of your paper.

1. Please describe methods in detail. For example Cell Viability Assay, Cell Death Assay , FACS-Based Lipid Peroxidation Assay. Add references wherever applicable.
2. Extraction of CoQ10- How is the purity. Give evidence
3. How did u dissolve cholesterol derived drugs. Please give evidence
4. Quantify all western blots and present them
- 5-Please add a diagram to show your findings

We thank the reviewers for professional and suggestive comments to help improve this manuscript. Below, please find our point-by-point responses to the editors and reviewers:

Reviewer #1

Major comments:

1. Figure 1a: The levels of lipid peroxidation should be provided to demonstrate that the cells were really protected from ferroptosis.

RESPONSE: we agree with the reviewer's comments and apologize for the lack of this piece of data. We have performed BODIPY 581/591-C11 staining and microscopy imaging after various sterols treatment (for both original and newly tested sterols). This data is now added as new Figure 1a-c and Figure s1a-c (line 150 and 152 in the manuscript).

Figure 1

Supplementary figure 1

2. Figure 2d, e: Whether the background lipid increase was corrected by FL2 channel when detecting lipid peroxidation using C11 BODIPY staining?

RESPONSE: we thank the reviewer for this critical question. We have indeed examined the red fluorescence of C11 BODIPY during our FACs analysis in Figure 2d, e. However, the original quantitative data was not corrected by FL2 channel (red fluorescence channel). We have now replaced this quantitative data after correcting FL2 channel as new Figure 2d, e (right panels) respectively.

Figure 2d

figure 2e

3. Figure S2e, f: The legend should be carefully checked. Cysteine starvation induced a nearly 90% of cell death (Figures S1e, S1h, S2c), while this was not observed in Figure S2e, f.

RESPONSE: we thank the reviewer for careful reading. We apologize for this carelessness. We carefully checked our lab note. In Figure s2c, cysteine starvation was 24h instead of 10h. In Figure s2e, f, cysteine starvation lasted for 17h instead of 27h. The legend was rewritten.

4. Figure 2a-c: Why did RSL3 induce a discrete frequency of ferroptosis.

RESPONSE: we appreciate the reviewer's comments and apologize for this confusion. As some of these experiments were carried out by different people in our lab, this discrete frequency of ferroptosis is mostly likely caused by different cell numbers that were initially seeded in culture plates in different experiments. As has been shown by Jiang lab (*Nature* **572**, 402–406 (2019)), cell interaction modulates sensitivity of cells to ferroptosis. We will control cell density better in the future.

5. Figures 3, 4: CH&Desmo also significantly attenuated ferroptosis of CoQ10-depleted or FSP1-deleted cells. Does that mean CH&Desmo inhibit ferroptosis independently of FSP1-CoQ10 axis? In Figure 4, FDFT1 deletion

alone was sufficient to abrogate the protective effect of CH on ferroptosis. The contribution of FSP1-CoQ10 axis remains doubtful.

RESPONSE: we really appreciate the reviewer for critical thinking. It also has been quite a confusion to us for a long time. We agree with the reviewer that partial attenuation of ferroptosis by CH&Desmo after CoQ10 depletion or FSP1 deletion could suggest either dependency or independency of CoQ10-FSP1 pathway in the action of CH&Desmo (original Figure 3). However, our hypothesis that CoQ10-FSP1 pathway plays a role in the action of CH&Desmo is supported by following data: 1. Inhibition of this pathway by either 4-CBA, shCOQ2 or knockout of FSP1 sensitizes cells to ferroptosis, suggesting that blocking this pathway exaggerated ferroptotic death under normal culture condition (Figure3 and 4). 2. We also performed new experiments to investigate whether exogenous CoQ10 could inhibit ferroptosis. Addition of 250nM exogenous CoQ10 led to around 25% increase of intracellular CoQ10 content as measured by mass-spectrometry assay (new Figure S3k). Importantly, it also reduced RSL3 induced cell death (new Figure 3k), suggesting that increasing cellular CoQ10 content can inhibit ferroptosis. 3. CH&desmo treatment increased cellular CoQ10 content by around 50%, 2folds more than that of after adding 250nM exogenous CoQ10 (Figure 3b). 4. Inhibition of FDFT1 by TAK-475 or knockout of FDFT1 has no effect on ferroptosis (Figure 4j, k), which could be explained by decreased levels of squalene but increased levels of CoQ, with a net "0" effect on ferroptosis.

We presented a diagram below to further explain our data in Figure 4 (numbers next to CoQ10 or squalene indicate relative cellular levels of these two molecules, 1 denotes basic levels under normal condition):

6. Appropriate statistical analysis should be used, such as one/two-way ANOVA and multiple comparison for three or more groups.

RESPONSE: we thank the reviewer for this critical point. We have made corrections. Please see below and new descriptions in the **Statistics and Reproducibility** section (line 764).

All comparisons were tested using unpaired two-tailed Student's t-test or one/two-way ANOVA with Tukey's post hoc test, with a confidence interval (CI) of 95%. $P \leq 0.05$ was considered statistically significant.

Minor comments:

1. Some grammatical mistakes such as "that involve" in Abstract, "array of" in Introduction, "are able to", "independent on each other" in Results and etc.

The manuscript should be thoroughly checked.

RESPONSE: We thank the reviewer for pointing out these grammatical mistakes. These mistakes were corrected.

2. Supplementary Figures particularly Figures S1 and S5 should be better cited and arranged to easily follow the results.

RESPONSE: We thank the reviewer for pointing out these issues. For Figure 1, original Figure s1c was moved to new Figure s1a, and original Figure s1a and s1b were moved to new Figure s1c, b respectively. We have added new citation for new Figure s1c (original Figure s1a) (citation 27). We also moved the text that describes Fig s1j-l after that of Figure s1g-l (line 171-179). For figure s5, we have moved original Figure s5b, c to Figure s5a, b respectively,

original s5d was moved to new Figure s5c, and original Figure s5a to new Figure s5d.

3. Figures 1, 5: What are Fer-1 and Lipo?

RESPONSE: we apologize for not including full names of these drugs. Fer-1 is Ferrostatin-1, and Lipo is Liproxstatin-1, we have changed Lipo to Lipro-1 in the manuscript. The full names were included where they first appeared in the text (second paragraph of **Results Section** for Fer-1 (line 167), second to the last paragraph of **Results Section** for Lipro-1 (line 364)).

Reviewer #2:

1. The data suggest that the levels cholesterol rather than its intermediates themselves eventually affect CoQ10 and squalene metabolism to inhibit ferroptosis. However, the intermediate metabolism seems to be focused in the manuscript and supplementary figures, so a graphic summary of ferroptosis inhibition by cholesterol would be helpful for readers.

RESPONSE: we thank the reviewer for this wonderful suggestion. We have added a graphic summary as new Figure 6.

Figure 6

Fig. 6. Schematic diagram of the mechanism of action of CH&Desmo. Under normal conditions, cells maintain basic levels of cellular squalene and CoQ10 to combat low extent of lipid peroxidation and ferroptosis. The balance can be easily disrupted by various stimuli to induce ferroptotic cell death. Under circumstances where cells are loaded with excess CH&Desmo, SQLE degradation is accelerated, which blocks carbon flux through squalene to cholesterol. Consequently, cellular levels of both squalene and CoQ10 are increased, and ferroptosis is inhibited.

2. Why several intermediates for cholesterol synthesis are dispensable for ferroptosis suppression? Did not they contribute to total cholesterol levels?

RESPONSE: we thank the reviewer's comments. To answer this question, we measured intracellular cholesterol levels by filipin staining followed by flow cytometry analysis in the presence of various sterol intermediates.

Desmosterol and 7-DHC indeed increased intracellular cholesterol, possibly because these two sterols are direct precursors to cholesterol (New Figure s2h), whereas other intermediates showed marginal or no effect on cellular cholesterol levels. However, inhibiting DHCR24 to block cholesterol biosynthesis from Desmosterol did not influence the effect of Desmosterol on ferroptosis inhibition (Figure 2f-h). As for 7-DHC, Angeli *et al* have shown that inactivation of DHCR7 that converts 7-DHC to cholesterol did not impact the anti-ferroptotic effect of 7-DHC (doi:10.21203/rs.3.rs-943221/v1 (2021)). (line 204-211)

New text added in the manuscript:

As upstream precursors to cholesterol, excess intermediates of cholesterol biosynthesis may enhance cholesterol biosynthesis, we thus tested this possibility by measuring intracellular cholesterol levels in the presence of sterol intermediates. Desmo and 7-DHC significantly increased cellular cholesterol levels as measured by flow cytometry, slightly less than that of CH supplementation. Latho also induced a marginal increase of intracellular cholesterol levels, whereas Zymo and Lano had no effect (Supplementary Fig. 2h).

3. If desmosterol inhibits ferroptosis via cholesterol synthesis, the inhibitory effect of desmosterol should be tested in DHCR7 KO cells

RESPONSE: we thank the reviewer for the suggestion. We have performed this experiment, and DHCR7 KO showed no effect on the function of Desmo. Please see new Figure 2l. (line 227)

4. In addition, as the authors suggest, the possible additional mechanism by which 7DHC inhibits ferroptosis via direct inhibition of autoxidation may be indicated in the figure. Furthermore, if possible, direct observation of 7DHC-mediated inhibition of autoxidation is required (Supp Fig. S3d, related to page 6).

RESPONSE: we thank for the reviewer's suggestion. We have performed the experiment as suggested, and 7-DHC indeed directly inhibited autoxidation.

(line 238). Please see new Figure s3d.

5. The levels of cholesterol in cells treated with MβCD should be provided. Because the direct observation of altered cholesterol levels upon avasimble and MβCD is critical, please provide these data (supplementary fig. 2d) in the main figure.

RESPONSE: we thank for the reviewer's suggestion. We have measured cholesterol content after MβCD treatment as suggested, and MβCD indeed reduced cholesterol levels by filipin staining. Please see new Figure s2d (right

panel). (line 194)

6. The involvement of CoQ10 in CH-mediated ferroptosis suppression is still

unclear. First, recycling of CoQ10 by FSP1 may be much more critical for ferroptosis suppression rather than the levels of CoQ10. Partial inhibition of ferroptosis by CH in FSP1 KO cells may suggest the requirement of FSP1 for CH-mediated ferroptosis suppression but it also imply that CH can also inhibit ferroptosis in the absence of FSP1. Could the author see the inhibition of ferroptosis by supplementing CoQ10 that can lead to 2 fold increase? This can be just discussed.

RESPONSE: we really appreciate the reviewer for critical thinking. We agree with the reviewer that partial attenuation of ferroptosis by CH&Desmo after CoQ10 depletion or FSP1 deletion could suggest either dependency or independency of CoQ10-FSP1 pathway in the action of CH&Desmo (original Figure 3). To address the issue whether the levels of CoQ10 is important, we performed new experiments to investigate whether exogenous CoQ10 could inhibit ferroptosis, as suggested by the reviewer. Indeed, addition of 250nM exogenous CoQ10 led to around 25% increase of intracellular CoQ10 content as measured by mass-spectrometry assay (new Figure S3k). Importantly, it also reduced RSL3 induced cell death (new Figure 3k), suggesting that increasing cellular CoQ10 content can inhibit ferroptosis.

Our hypothesis that CoQ10-FSP1 pathway plays a role in the action of CH&Desmo could further be supported by the result that inhibition of FDFT1 by TAK-475 or knockout of FDFT1 has no effect on ferroptosis (Figure 4j, k), which could be explained by decreased levels of squalene but increased levels of CoQ, with a net “0” effect on ferroptosis.

We presented a diagram below to further explain our hypothesis (numbers next to CoQ10 or squalene indicate relative cellular levels of these two molecules, 1 denotes basic levels under normal condition):

7. Fig. 5a, Lipo may be corrected to Lipro-1.

RESPONSE: we thank the reviewer for pointing out this mistake. Correction was made in the new version of manuscript.

Reviewer #3:

1. Please describe methods in detail. For example Cell Viability Assay, Cell Death Assay, FACS-Based Lipid Peroxidation Assay. Add references wherever applicable.

RESPONSE: we appreciate the reviewer's suggestion. We have described methods in more detail and added more references. (line 570-631)

2. Extraction of CoQ10- How is the purity. Give evidence

RESPONSE: we thank the reviewer for comments. We apologize that the purity was not examined in our experiment. The protocol we used for CoQ10 extraction as adapted from Doll *et al* (Nature, 575, 693–698 (2019)). In this protocol, CoQ levels were measured by LC-MS after extraction, the same amount of CoQ9 was added in each sample as internal control, and commercial CoQ10 was used as a standard during LC-MS. Thus, this well controlled assay rendered the purity of CoQ10 not so critical. In another report, similar method was also used to quantify cellular CoQ10 levels (Nature 575, 688–692 (2019)).

3. How did u dissolve cholesterol derived drugs. Please give evidence

RESPONSE: we thank the reviewer for pointing this out. All cholesterol derived drugs were dissolved with M β CD according to the method described by Ruthellen et al (Cell Reports, 37(7), 2021). We have added details and cited this paper in **Materials and Methods Section**. (Line 570-577)

4. Quantify all western blots and present them

RESPONSE: we thank the reviewer for this suggestion. WB results were quantified as suggested.

5-Please add a diagram to show your findings

RESPONSE: we thank the reviewer for this suggestion. We have added this diagram as new figure 6.

REVIEWERS' COMMENTS:

Reviewer #1 (Remarks to the Author):

The authors have addressed all my concerns.

Reviewer #2 (Remarks to the Author):

The authors have fully addressed all the issues raised.

Reviewer #3 (Remarks to the Author):

Accept